# The Physiochemical Properties and Adsorption Characteristics of Processed Pomelo Peel as a Carrier for Epigallocatechin-3-Gallate

**DOI:** 10.3390/molecules25184249

**Published:** 2020-09-16

**Authors:** Liangyu Wu, Guoying Zhang, Jinke Lin

**Affiliations:** 1College of Horticulture, Fujian Agriculture and Forestry University, Fuzhou 350002, China; 000q021014@fafu.edu.cn (L.W.); 2160305002@fafu.edu.cn (G.Z.); 2College of Anxi Tea, Fujian Agriculture and Forestry University, Quanzhou 362406, China

**Keywords:** processed pomelo peel, EGCG, isothermal adsorption, physiochemical characteristics, thermodynamics

## Abstract

The NaOH-HCl- and ethanol-pretreated pomelo peel samples were prepared to apply to the batch adsorption for epigallocatechin-3-gallate (EGCG). The characteristics of peel samples were determined by Fourier transform infrared spectroscopy, scanning electron microscopy and a laser particle analyzer. The results of the physiochemical properties of the peel samples demonstrate that these peel samples have a promising adsorption capacity for EGCG, because of the increased potential binding sites on the surface compared with those of untreated peel samples. These two peel samples showed enhanced adsorption capacities of EGCG compared with that of unmodified peel in terms of the isothermal adsorption process, which could be described by both Langmuir and Freundlich models, with the theoretical maximum adsorption capacity of 77.52 and 94.34 mg g^−1^ for the NaOH-HCl and ethanol-treated peel samples, respectively. The adsorption kinetics demonstrated an excellent fitness to pseudo-second-order, showing that chemisorption was the rate-limiting step. The thermodynamics analysis revealed that the adsorption reaction was a spontaneous and endothermic process. This work highlights that the processed pomelo peels have outstanding adsorption capacities for EGCG, which could be promising candidates for EGCG delivering in functional food application.

## 1. Introduction

Tea catechins are a class of abundant polyphenolic compounds belonging to flavanols present in *Camellia sinensis* (L.), of which epigallocatechin-3-gallate (EGCG) is the most important component with the highest content and several health benefits, including anti-irradiation [1], anti-carcinogenic [2], anti-inflammatory [3], anti-oxidative [4] and anti-cardiovascular [5] features, etc. However, the stability and bioavailability of EGCG are quite susceptible to oxidants, high temperatures and alkaline environments; thus, the degradation or epimerization could be observed when EGCG has suffered from those adverse factors [6,7,8,9], which limits the application of EGCG in functional foods.

The design of appropriate carriers is a crucial approach to improve the bioavailability of EGCG. The lipid nanoparticlulate delivery system has been used for EGCG delivering, which protected EGCG against degradation during gastrointestinal digestion, with the stability of encapsulated EGCG ranging from 80% to 90% [10]. Whey protein has also been applied to the formation of the EGCG complex to stabilize the EGCG in digestion simulation, and the phenolic showed a more stable property when associated with the protein [11]. The controlled-release rate of EGCG from zein/chitosan nanoparticles in 95% ethanol fatty simulant could arrived at 32%, with a long-term protection against oxidation for fatty foods [12]. Other matrixes for EGCG carrying include rice bran [13] and β-glucan [14], with the predicted maximum adsorption amounts of 112.36 and 333.33 μg mg^−1^, respectively. 

Recently, there is increasing interesting on the edible by-products from agriculture products or food processing recently, as they are underutilized and could be reutilized in the food cycle to reduce the food waste or environmental pollution. Pomelo (*Citrus grandis*) peel, as the byproduct of the juice extraction processing industry, is a low-cost food industrial waste that is readily available in large quantities. The yield of pomelo peel is nearly to 4.22 million tons in 2018; however, most of the peel was disposed as waste, resulting in severe environmental issues [15]. Compositional data showed that pomelo peel is rich in dietary fibres (~42%), and the content of cellulose is around 50%, the particle of pomelo peel consisted of porous microstructures with a high specific surface area, which make pomelo peel a promising carrier for phytonutrients [16,17]. Our previous work confirmed that pomelo peel was a useful matrix for binding EGCG; however, the theoretical maximum monolayer adsorption capacities of EGCG onto pomelo peel-based adsorbents were still less than 30 mg g^−1^ dry mass at 25–55 °C [16]. 

The alkaline-acid elution has been applied to the pretreatment of various plant-based adsorbents, in order to enhance binding sites by removing soluble substances [18,19]; washing with ethanol to increase the surface area to prepared modified adsorbent has also been reported in previous study [20]. In this study, NaOH-HCl- and ethanol-modified pomelo peel samples were prepared and then subjected to the adsorption for EGCG. The physicochemical features of the peel samples were determined by Fourier transform infrared spectroscopy (FTIR), scanning electron microscopy (SEM) and laser particle analyzer. The adsorbing performance of EGCG onto the pretreated pomelo peel in aqueous solutions were fitted to adsorption kinetics, isothermal and thermodynamic models to evaluate the interactions between the pomelo peel and EGCG.

## 2. Results and Discussion

### 2.1. The Properties of Pomelo Peel

The FTIR spectra (Figure 1a) presented the alteration in the functional groups of NaOH-HCl modified pomelo peel (NHP) sample and ethanol-modified peel (EP) samples in comparison with unprocessed pomelo peel (PP) in our previous study [16]. Among all the three peel samples, the broad bands around 3385 cm^−1^ in peel samples were ascribed to the hydroxyl groups or intermolecular hydrogen bonding from lignocellulose [21], and the band observed around 2918 cm^−1^ was caused by the stretching vibration of C-H in methyl or methylene groups [22]. The band appeared around 1740 cm^−1^ in PP was attributed to the stretching vibration of the C=O bond from non-ionic carboxyl groups (-COOCH_3_, -COOH), which was also observed in EP peel, while the decrease of this band in the NHP peel sample implied that the phenolic acids were probably removed from the NHP after NaOH-HCl elution, in agreement with the previous report in NaOH-HCl-treated apple pomace [21]. The band peaked at 1645 cm^−1^ in the NHP sample could be related to C=O vibration of acid or ester, indicating alkaline treatment induced a saponification effect, resulting in a asymmetric stretching vibrations increasing of ionic carboxylic acid groups (-COO^−^), while this band was shifted to 1633 cm^−1^ in EP with a little decrease, suggesting that the carboxylic functional group was probably esterified by ethanol [23]. The adsorption band at 1616 cm^−1^ in PP was associated with the stretching vibration of carbonyl-carbonyl [15]. The bands ranged from 1424 to 1370 cm^−1^ in all the three peel samples (Figure 1a) and were assigned to the scissoring vibration of methylene and in-plane rocking or bending vibration of methyl, in line with the characteristic signs of lignocellulose in plant-based adsorbents [21,24]. The band at 1264 cm^−1^ originated from the vibration of Ar-O in lignin [7], and the band at 1240 cm^−1^ in EP derived from the C-O stretching in carboxylic acid of hemicellulose [25]. The absorbance at 1318 cm^−1^ in NHP represented the O-H bending in cellulose [26], and the weak shoulder at 1160 cm^−1^ was corresponding to the arabinosyl side chains of lignocellulose [27]. The results of the FTIR spectra indicate that the functional groups’ changes resulted from pretreatments, especially the alkaline–acid elution, during which some soluble compounds such as pectin or protein might have been removed by the NaOH or HCl solution, with most insoluble leftovers (cellulose, lignin) retained in the biosorbent, in agreement with the observation of citric acid and sodium hydrogen carbonate treated plant-based adsorbent [28]. 

The FTIR spectra for pretreated peel samples before and after adsorption were also presented in Figure 1a. For the peel samples adsorbed with EGCG, the absorbance around 3385–3387 cm^−1^ in NHP and EP were shifted to 3360–3363 cm^−1^ after adsorption, with a little decrease upon interaction with EGCG (Figure 1a). The NHP and EP peel sample had C=O functional group peaks located around 1645 and 1738 cm^−1^ were shifted and weakened after EGCG adsorption, indicating that interactions took place at these functional groups. On the whole, the band intensities were decreased after the adsorption of EGCG onto NHP and EP samples, suggesting that functional groups on the surface of peel samples contributed to EGCG binding. The FTIR spectra demonstrated that there are significant shifts at -OH, C-C, and C=O group bands between the peel samples before and after adsorption, implying that EGCG binding could mostly be at -OH and C=O groups. Lots of evidence has showed that polyphenols mainly interact with various carbohydrates derived from the cell wall in the plant-based matrix including cellulose, pectin or lignin [28,29,30,31]. Hydrogen bonds are formed among hydroxyl groups in galloyl group of EGCG and oxygen atoms of the glycosidic linkages in the polysaccharides; except for hydrogen bonding, hydrophobic interactions between the benzene rings or alkanes of lignocellulose and phenolic rings of tea catechins were also contributed to the adsorption of polyphenols onto biosorbents [14,32].

The particle size distributions of the NHP, EP and PP samples are presented in Figure 1b. The particle size of the NHP sample, with the average volume diameter (D[4,3]) of 589.2 μm and the d (0.9) of 1179.9 μm, was markedly larger than those of the EP sample (D[4,3]: 150.3 μm, d (0.9): 391.0μm). As shown in Figure 1b, the particle size of NHP sample demonstrated an approximate normal distribution around 630.9 μm, with a peak volume of 7.8%. By contrast, there was a skewed distribution in the particle size constitution of the EP sample, with two peaks observed at 79.4 and 316.2 μm, respectively, which was similar with the particle distribution of untreated pomelo peel with two peaks observed around 101.1 and 309.6 μm (Figure 1b). The bimodal curve of particle size distribution in pomelo peel was attributed to the separated cell clusters produced during the milling and thermal processing [33]. The similar particle size distributions observed in PP and EP samples suggested that the ethanol treatment retained most of the peel fragments around 79.4 and 316.2 μm, without significant changes compared with untreated peel. The strong-polarity solution was able to decompose the amorphous region consisting of lignocellulose in plant-based materials and remove soluble polysaccharide, contributing to a redistribution of the fragments [17,26,34], which is consistent with the observation that a large cell cluster was kept in the NHP peel after the removal of some small soluble fragments by NaOH-HCl elution.

The images of a scanning electron microscope (SEM) on the ultrastructure of PP, NHP and EP samples are displayed in Figure 2. The images revealed that the smooth flaky surface and little porous structure were observed in PP sample, which had a tenuous effect on the adsorption capacity. As shown in Figure 2c,d, the morphology of NHP peel presented compact and rough surface with visible wrinkles with further amplification (Figure 2d). In contrast, a rougher and more porous structure was observed in the micrographs of EP (Figure 2e,f) compared with that of NHP. After the alkaline–acid pretreatment, the cellulose, lignin and a few pectin were separated [26,35], resulting in the disruption of the cell wall layer, with some of the cracks and porous structure appearing. The pretreatment of ethanol could create a bigger pore size than NaOH/HCl elution in the plant-based biosorbent, potentially providing enhanced adsorption capacity by retaining cellulose, hemicellulose, and lignin [26,36].

### 2.2. Adsorption Kinetics Analysis

The pH of EGCG-pomelo peel adsorption system was maintained around 3.8–4.0 over the adsorption process. As shown in Figure 3a, the adsorption of EGCG onto these two peel samples showed a logarithmic growth tendency over the adsorption process. The adsorption capacity (q_t_) raised rapidly within 120 min, approaching nearly 50% of the equilibrium adsorption capacities of the two samples, subsequently slowed down into the plateau phase (240–720 min), and finally arriving at equilibrium points at 1440 min. During the initial phase of adsorption process, there were amounts of available adsorption sites on the surface of these peel samples, resulting in a radical increasing in the adsorption capacities, then the adsorption capacity decreased, which was attributed to the sharp reduction in adsorption sites and the enhanced repulsive forces of EGCG between those adsorbed on the peel and those that remained in solution phase. Moreover, Figure 3a demonstrates that the equilibrium adsorption capacity of EGCG onto EP (67.66 mg g^−1^) was better than that of NHP (52.83 mg g^−1^), which was likely caused by the difference in the internal microstructures of the two samples.

The experimental data were fitted to pseudo-first-order and pseudo-second-order models so as to describe the adsorption kinetics of EGCG onto the processed pomelo peel (Figure 3b,c). The parameters calculated from these two models are presented in Table 1. With higher correlation coefficients (R^2^) and smaller relative errors (Table 1), the equilibrium adsorption capacities (q_e.cal_) predicted from the pseudo-second-order model were much closer to the experimental data (q_e.exp_) than those from the pseudo-first-order model, which is similar to the previous study on modified biosorbents adsorbing organic compounds [37]. The fact that the well-fitness of our experimental data to pseudo-second-order indicated that the rate-limiting step was the chemisorption process, in which the valence forces through the sharing or exchange of electrons exerted between the adsorbent and adsorbate [38].

### 2.3. Adsorption Isotherm Analysis

The Langmuir and Freundlich isothermal models were applied to describe the relationship between the adsorption capacity of adsorbent (q_e_) and the concentration of the adsorbates (C_e_) at equilibrium at a given temperature. The plots of the equilibrium concentration of EGCG and adsorption capacity of processed pomelo peel at different temperatures are presented in Figure 4, in comparison with the data for unprocessed pomelo peel (PP) obtained previously [16]. As shown in Figure 4, the qe values increased with an increase in the initial EGCG concentration from 50 to 800 mg L^−1^, displaying a logarithmic growth over the isothermal adsorption process. This phenomena was attributed to the following reasons: firstly, the adsorption process took place in an efficient way when the available adsorbing sites on the surface of the peel were surrounded by massive EGCG molecules; secondly, the increasing initial concentration of EGCG intensified the repulsive force between the EGCG on adsorbent and EGCG remained in solution, leading to the slow rise in adsorption capacity observed during isothermal adsorption (Figure 4). Notably, the q_e_ values observed in NHP and EP samples were clearly higher than that of unmodified pomelo peel ranging from 25 to 55 °C (Figure 4), indicating that the alkaline–acid and ethanol pretreatments markedly improve the adsorption characteristic of pomelo peel by enhancing the available adsorption sites on the surface of pomelo peel [39].

The Langmuir and Freundlich isothermal models were applied to describe the monolayer and multilayer adsorption onto the adsorbent at given temperature, respectively. In this study, the experimental data were fitted to these models, and the fitting plots are demonstrated in Figure 5. The results show that the isothermal adsorption behavior of EGCG onto these two peel samples were able to be described by these two models with all the correlation coefficients (R^2^) higher than 0.90 (Table 2 and Table 3). The theoretical maximum monolayer adsorption capacities (Q_m_) of EGCG onto NHP and EP were found to be 77.52 and 94.34 mg g^−1^ at 25 °C, respectively (Table 2), all of which are significantly higher than that of unmodified pomelo peel (29.24 mg g^−1^) in our previous work [16]. Additionally, the theoretical maximum adsorption capacities of processed peel samples showed a decreasing trend along with the rise in ambient temperature (Table 2), suggesting that a low temperature was favorable to the adsorption of EGCG onto processed pomelo peel. It is postulated that the EGCG molecules were driven to escape from the surface of pomelo peels at high temperature, resulting in a low adsorption capacity, which was also been reported in other biomass materials [40].

In the Freundlich model, the K_F_ value is an index to describe adsorption capacity. The minimum values of K_F_ and 1/n were obtained at 55 °C, and the maximum values were obtained at 25 °C in this study (Table 3). The variation trends of K_F_ paralleled with those of Q_m_ predicted from the Langmuir model, showing a consistency in this study. The 1/n is a parameter related to the favorability level of the adsorption process. In the present work, the values of 1/n for NPH was lower than 0.5, while the values for EP were higher than 0.5 (Table 3), revealing that the adsorption process of EGCG onto NHP and EP were favorable and pseudo-linear, respectively [41].

Different modification methods including physical or chemical pretreatments have been used to enhance the adsorption capacity of adsorbents. The NaOH elution would remove the soluble polysaccharides in the adsorbent, and HCl-treatment was found to boost the adsorption capacity by decreasing ash and the protonation of functional groups (-COO^−^ and -O^−^), supplying more effective binding sites [34]. The application of ethanol to the pretreatment is able to maximum the biosorption capacity via esterification of carboxyl groups and dissolvent of some organic compounds in lignocellulose materials [26]. In the present study, two different pretreatments were used, leading to different reactions between functional groups and adsorbing surface of pomelo peel samples, obtaining the biosobents with better sorption efficiency and capacity compared with untreated peel. While the ethanol-treated peel sample possessed a surface with more porous structures and a smaller particle size distribution than those of the NaOH-HCl-treated sample (Figure 1 and Figure 2), which could provide more activity binding sites for EGCG molecules. Several studies have also used alcohols to improve the adsorbing features of biosorbents, in which the major parts of lignocellulose were still maintained with the promotion of effective adsorbing surface area [20,42,43].

In this study, temperature was a dominant factor affecting the adsorption capacity of EGCG onto peel samples. The analysis on adsorption capacities showed that both of NHP and EP have higher adsorbing amounts at 25 °C than at 40 and 55 °C (Figure 4). Depending on the molecular interactions, the adsorption of EGCG onto peel samples was mainly driven by the cooperative hydrogen bonding between the hydroxyl groups of EGCG and oxygen atoms of polysaccharides as mentioned previously. Generally, the enthalpy of hydrogen bond formation is negative, meaning the inter-association equilibrium constant and the number of the hydrogen bonds will decrease upon increasing the temperature [44]. As reported, the hydrogen bonds are formed between tea polyphenols and β-glucan [45] polyvinylpolypyrrolidone [46], saw dust [47], during which these absorbents arrived at the maximum adsorption capacities around 15–30 °C. Except for the chemical adsorption process, increasing temperature both elevate the solubility of polyphenols in water and the thermal motion of polyphenol molecules, hindering the binding of polyphenol onto polysaccharide [48]. Taken together, a relatively low temperature is beneficial to the adsorption of EGCG onto biosorbents based on the results of previous studies and present work.

### 2.4. Adsorption Thermodynamics Analysis

The standard Gibbs free energy change (ΔG°), enthalpy change (ΔH°), and entropy change (ΔS°) can be used to describe the spontaneous nature, endothermic nature and randomness of the adsorption process, respectively. In present work, the ΔH° of EGCG adsorption onto the NHP and EP pomelo peel samples were 21.50 and 27.31 kJ mol^−1^, respectively (Table 4), indicating that the adsorption was an endothermic process; the ΔS° for these two samples were 147.14 and 156.90 J mol^−1^ K^−1^ respectively, suggesting that the randomness of the EGCG-pomelo peel adsorption system was increased according to the positive value of ΔS°. All the values of ΔG° calculated in this study were negative, revealing that the adsorption reaction of EGCG onto the modified pomelo peel took placed spontaneously. The adsorption process could be classified into chemisorption with ΔG° ranges from −400 to −80 kJ mol^−1^ and physisorption with ΔG° ranges from −20 to 0 kJ mol^−1^ according to Jaycock and Parfitt [49]. The fact that ΔG° ranged from −19.48 to −26.98 kJ mol^−1^ in this study (Table 4) illustrated that the adsorption process of EGCG onto processed pomelo peel was likely dominated by both physisorption and a chemisorption mechanism. The proposed mechanism of the interactions between polyphenols and plant-based polysaccharides mainly involves noncovalent binding, such as hydrogen bonding, hydrophobic forces as well as ionic interactions [50], the aromatic rings in polyphenols and sugar rings in polysaccharides is approaching upon the formation of hydrogen boding, allowing the generation of Van der Waals force [51], resulting in the joint dominance over the binding of polyphenols with polysaccharides. Therefore, the adsorption thermodynamics analysis in this study consolidates that the hydrogen bonding may be partly responsible for the binding between EGCG and plant-based adsorbents, while Van der Waals interactions are also involved in the adsorption process. Moreover, the value of ΔG° observed in our study is of a similar magnitude to that for EGCG adsorbing to unmodified pomelo peel (ranging from −19.46 to −22.10 kJ mol^−1^) [16], implying that the adsorption reaction is independent of the temperature over the adsorbing process, the modification treatments only increased the available adsorbing sites on pomelo peel through eluting soluble components, instead of altering the adsorption mechanism.

### 2.5. Desorption of EGCG from the Pretreated Pomelo Peels

After the three-stage elution, both of the total desorption rates of EGCG from NHP and EP were higher than 80%, with negligible difference less than 2% (Table 5). Remarkably, the desorption rates in ethanol eluate (the 2nd and 3rd elution stage) were much higher than those in water eluate (the 1st elution stage). The high desorption efficiency of EGCG from pretreated peels by ethanol solution might be attributed to the disturbance effect of ethanol on the hydrogen bond interactions and hydrophobic associations between lignocellulose and EGCG as an interfering reagent [52,53]. A similar result has also been reported in the application of lignocellulose-enriched tea stalk as an adsorbent in binding catechins [54]. The result of the desorption test suggested that the complex consisted of EGCG and pretreated pomelo peel samples could be stable in water, which could be candidates for the application of EGCG delivery to enhance the bioavailability in vivo. However, the adverse environmental pH and enzymes in gastric and intestinal digestion fluids are the major factors that induce the degradation of EGCG, the in vitro or in vivo digestion tests are needed to valid the feasibility of pretreated pomelo peels as carriers for EGCG.

## 3. Materials and Methods

### 3.1. Materials and Reagents

EGCG (≥95%) was obtained from Huzhou Sifeng Biochem Co., Ltd. (Huzhou, China). The HPLC reference compounds were supplied by Sigma-Aldrich (Darmstadt, Germany). All other chemical reagents (Sinopharm chemical reagent Co., Ltd., Shanghai, China) were of analytical purity, and Milli-Q water was supplied by Yingke Water System Group (Fuzhou, China).

The pomelo peel was purchased from a local market in Fuzhou, China. The peel was milled by a colloid mill (FW177, Taisite Instrument Co., Tianjin, China) just after arriving at the laboratory, then the milled sample (wet peel) was obtained. The wet peel was subjected to the followed treatments: (i) the peel was drenched in four volumes of 0.1 M NaOH for 72 h at room temperature, then the solution was drained, and the peel sample was washed with Milli-Q water to neutrality; the alkaline-treated sample was subsequently drenched in four volumes of 0.1 M HCl for another 72 h at room temperature, then washed with Milli-Q water to neutrality, then the NaOH-HCl modified pomelo peel (NHP) sample was obtained; (ii) the wet peel sample was drenched in four volumes of ethanol, stirred at room temperature for 72 h at room temperature, then the ethanol solution was drained, and the peel sample was washed with Milli-Q water to neutrality, the ethanol modified pomelo peel (EP) sample was obtained. All the peel samples were freeze dried and storage at −20 °C for further use.

### 3.2. Characterization of Pomelo Peel Samples

The functional groups in the processed peel samples were identified on a Nicolet Avatar 370 FTIR spectrometer (Thermo Fisher, Waltham, MA, USA) with wavelengths that ranged from 4000 to 500 cm^−1^ according to the instruction manual. The ultrastructure of sample was observed using a XL-30-E scanning electron microscope (Philips, Amsterdam, Netherlands). The particle size distribution of the peel samples were determined on a laser diffraction analyzer (Mastersizer 3000, Malvern Instruments Ltd., Malvern, UK).

### 3.3. Adsorption Experiments Study

The different pretreated pomelo peel samples were taken out from frozen storage and rinsed with Milli-Q water, then filtered with Whatman No. 3 filtering paper. The moisture content of the rinsed samples was measured by a MA-150 moisture-meter (Group of Sartorius, Göttingen, Germany) prior to adsorption study. The pH of the adsorption system was determined by a pH meter (FiveEasy, Mettler Toledo, Zurich, Switzerland).

### 3.4. Adsorption Kinetics

A series of rinsed pomelo peel samples (two grams for each) were put into conical flasks containing 200 mL EGCG aqueous solution with an initial concentration of 200 mg L^−1^. Then, the conical flasks were sealed by parafilm (Pechiney Plastic Packaging Inc., Chicago, IL, USA) and shaken (60 rpm) in a water bath at 25 °C. One milliliter supernatant was sampled from each flask at specific time intervals (5, 10, 15, 30, 60, 120, 240, 720 and 1440 min) after initial adsorption, then the supernatant samples were centrifuged at 12,000 rpm at 5 °C for 15 min, and subjected to HPLC determination. The amount of EGCG adsorbed onto pomelo peel at time t, q_t_ (mg/g) was calculated by Equation (1):(1)$$q_{t}=\frac{(C_{0}-C_{t})V}{W}$$
where q_t_ was the adsorption capacity (mg g^−1^) of EGCG onto pomelo peel at time t. C_0_ and C_t_ (mg L^−1^) were the concentration of initial EGCG and free EGCG retained in the liquid phase at time t, respectively. V (L) was the volume of the solution at time t and W (g) is the dry mass of pomelo peel used in the study.

The experimental data were fitted to pseudo-first-order (Equation (2)) and pseudo-second-order (Equation (3)), respectively, in order to describe the mechanism of adsorption process.
(2)$$\ln(q_{e}-q_{t})=\ln q_{e}-k_{1}t$$
(3)$$\frac{t}{q_{t}}=\frac{1}{k_{2}q_{e}{}^{2}}+\frac{t}{q_{e}}$$
where q_e_ and q_t_ (mg g^−1^) are the adsorption capacities of EGCG adsorbed onto pomelo peel at equilibrium and time t (min), respectively; k_1_ (min^−1^) and k_2_ (g mg^−1^ min^−1^) are the constants of pseudo-first-order and pseudo-second-order, respectively.

### 3.5. Isothermal Adsorption Study

Samples of rinsed pomelo peel (2.0 g) were added to a series of conical flasks containing 200 mL of EGCG solution with initial concentrations of 50, 100, 200, 400 and 800 mg L^−1^. Subsequently, all the flasks were covered with parafilm and shaken (60 rpm) at a water bath at 25, 40 and 55 °C for 480 min. An aliquot of the sample (1 mL) from each flask was taken and centrifuged at 12,000 rpm and 5 °C for 15 min. The supernatant samples were subject to HPLC analysis to determine the concentration of free EGCG at equilibrium. The Langmuir (Equation (4)) and Freundlich (Equation (5)) isotherm models were applied to fit the experimental data, to describe the isothermal adsorption process.
(4)$$\frac{C_{e}}{q_{e}}=\frac{1}{Q_{m}K_{L}}+\frac{C_{e}}{Q_{m}}$$
where C_e_ (mg L^−1^) is the free EGCG concentration at equilibrium in liquid phase and q_e_ (mg g^−1^) is the adsorption capacity of EGCG onto pomelo peel at equilibrium. The parameter Q_m_ (mg g^−1^) is the theoretical maximum adsorption capacity, and K_L_ is the Langmuir equilibrium constant; these two parameters can be calculated from the slope and intercept of plot of C_e_/q_e_ versus C_e_ in this study.
(5)$$\ln q_{e}=\frac{1}{n}\ln C_{e}+\ln K_{F}$$
where 1/n is the Freundlich constant associated to the favorable level of the adsorption process and K_F_ is the constant relative to the adsorption capacity of the adsorbent.

### 3.6. Thermodynamic Analysis

Thermodynamic parameters, including the standard Gibbs free energy change (ΔG°), enthalpy change (ΔH°), and entropy change (ΔS°), were employed to evaluate the spontaneous, endothermic nature and randomness change of the adsorption reaction. ΔG° can be obtained according to Equation (6):(6)$$\Delta G{}^{\circ}=-RT\ln K_{a}$$
where R is universal gas constant (8.314 J mol^−1^ K^−1^), T is the thermodynamic temperature in Kelvin. K_a_ is the thermodynamic equilibrium constant, which can be replaced by Langmuir equilibrium constant (K_L_) when adsorbates were with weak charges (e.g., organic compounds) during the adsorption process [55]. Hence, Equation (6) could be stated as Equation (7):(7)$$\Delta G{}^{\circ}=-RT\ln K_{L}$$

The ΔH° and ΔS° can be predicted from the slope and intercept of Van’t Hoff equation (Equation (8)), which could be transformed to Equation (9).
(8)ΔG°=ΔH°−TΔS°
(9)$$\ln K_{L}=\;\Delta S{}^{\circ}∕R-\;\Delta H{}^{\circ}∕RT$$

### 3.7. Desorption Test

The desorption test of static elution was performed according to the previous study [46] with a minor modification: the MillQ water rinsed pretreated peel sample (2.0 g) was mixed with 200 mL of EGCG solution (800 mg L^−1^) and shaken at 25 °C in a waterbath (60 rpm) for 480 min. The peel sample was collected after adsorption on a Buchner funnel by vacuum filtration, then incubated with Milli-Q water (100 mL) at 25 °C, 150 r min^−1^ for 60 min at the first elution stage. Then, the peel sample was collected again by vacuum filtration, and further incubated with 50 mL of ethanol (70%, *v/v*) for 60 min at the second elution stage. After filtration and collection, the third elution was conducted by incubating the peel samples with 50 mL of ethanol (70%, *v/v*). The volume of the three separated eluates was measured. The concentration of EGCG in adsorption solutions (before and after) and separated eluates from the first, second and third elution stages were subject to HPLC determination.

### 3.8. HPLC Determination

The determination of EGCG concentration was according to the method used in our previous studies [8,40] with a little modification. Briefly, the HPLC conditions were as follows: column temperature, 26 °C; injection volume 10 μL, VYDAC C18 monomeric column (250 × 2.1 mm inner diameter × 5 μm particle size), mobile phase A, acetonitrile + 0.1% trifluoroacetic acid; mobile phase B, water + 0.1% trifluoroacetic acid; linear gradient elution, from 3% A/97% B to 25% A/75% B within 40 min and then changed to 3% A/97% B until 45 min, with 200 μL/min flow rate. The ultraviolet-visible detector wavelength was set at 275 nm. To quantify EGCG, the detected peak areas were compared with that of the external EGCG standard solution. There was no EGCG peak detected when an extract from the modified pomelo peel alone was subjected to the HPLC analysis, confirming that there is no EGCG existing in pomelo peel samples used in the present study. When the extract from modified pomelo peel samples alone was subjected to HPLC analysis, there was no EGCG peak detected at the expected retention time, confirming the absence of EGCG in the pomelo peel samples used in this study.

## 4. Conclusions

There has been increasing interest in applying edible biosorbents to delivering phytonutrients from aqueous solutions. In particular, surface modification with various eluents is a universal approach to improve the adsorption capacity of many plant-based materials. In this study, the pomelo peel samples with NaOH-HCl and ethanol pretreatments were used to compare the adsorption capacity of EGCG with that of untreated peel. The adsorption kinetics analysis demonstrated that the adsorption process had a better fitness to the pseudo-second-order, and both Langmuir and Freundlich isothermal models were adequate to describe the relationship between the EGCG concentration and adsorption capacities of pomleo peel at a constant temperature. More importantly, the NHP and EP peel samples showed better theoretical maximum adsorption capacities (77.52 and 94.34 mg g^−1^) than that of untreated peel (29.24 mg g^−1^), suggesting that ethanol-treated pomelo peel could be a promising carrier for the delivery of EGCG considering the operating convenience of pretreatment in this study. However, it is also important to investigate the bioavailability of EGCG-pomelo peel complex in vivo to determine whether this can be translated to the achievement of desirable physiological properties in further study.

## Figures and Tables

**Figure 1 molecules-25-04249-f001:**
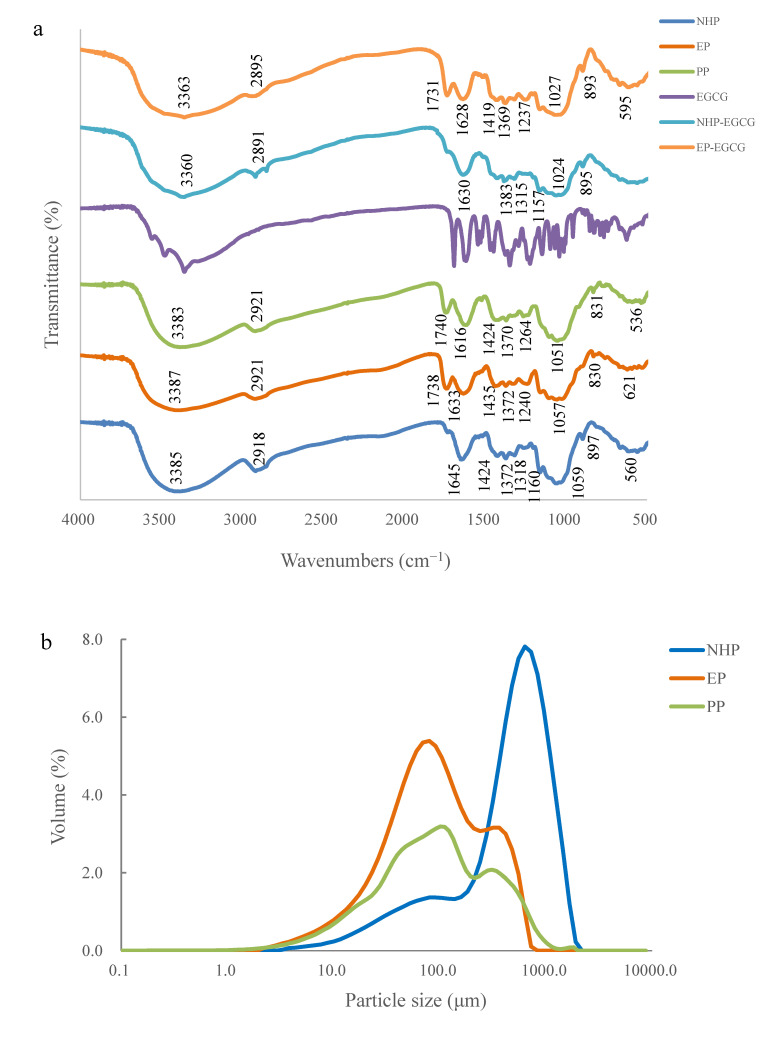
FTIR spectra of PP, NHP, EP pomelo peel samples, EGCG powder, and NHP, EP after adsorption (**a**). The particle size distribution of NHP, EP and PP peel sample (**b**). The curves of untreated pomelo peel (PP) in FTIR and particel size distribution results were re-drawn from our previous result in reference [16].

**Figure 2 molecules-25-04249-f002:**
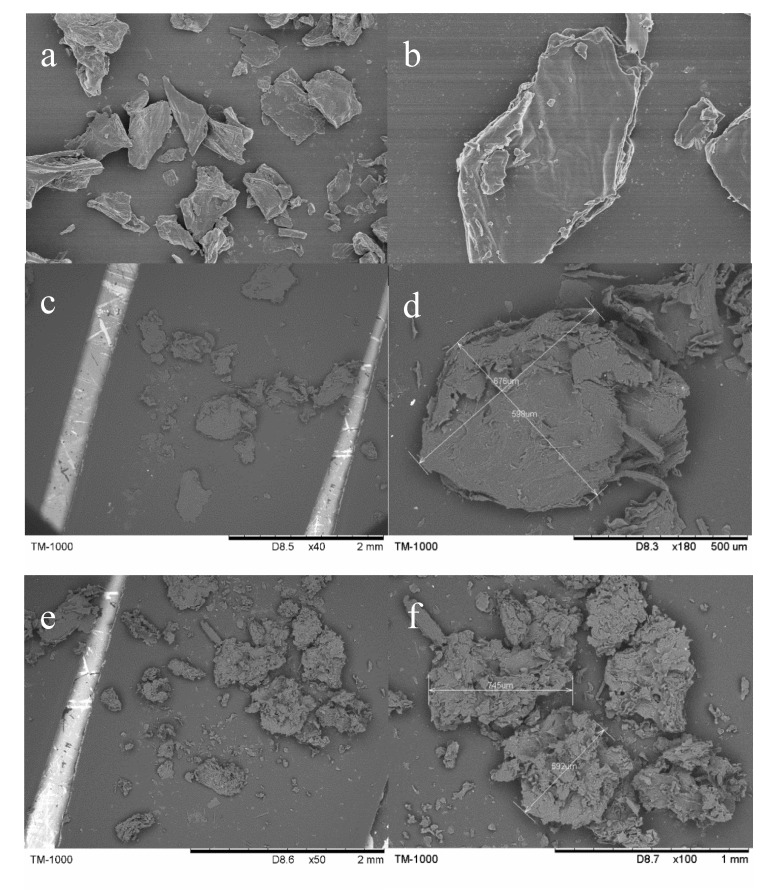
SEM images of PP (**a**,**b**), NHP (**c**,**d**) and EP (**e**,**f**) samples of pomelo peel.

**Figure 3 molecules-25-04249-f003:**
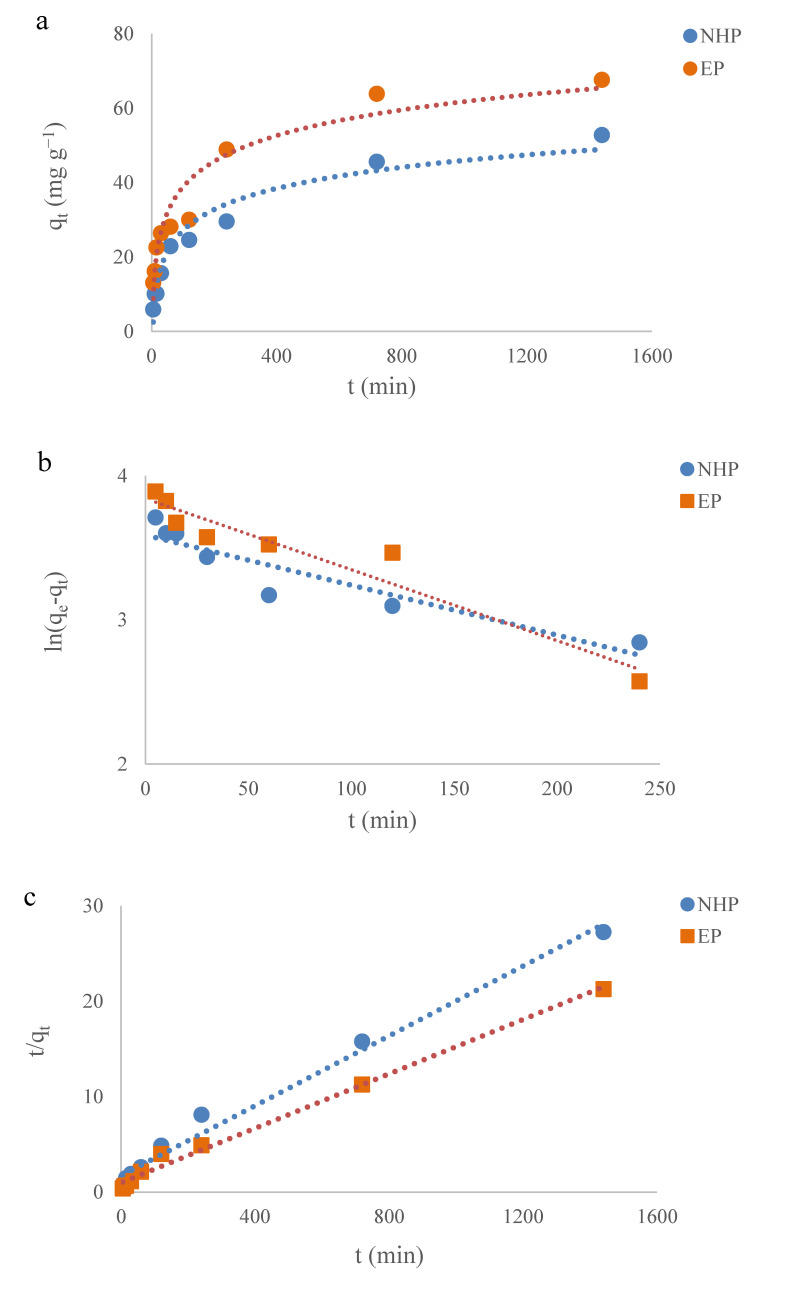
Adsorption kinetics of EGCG onto processed pomelo peel (**a**), and the experimental data fitted to pseudo-first-order (**b**) and pseudo-second-order models (**c**), respectively, at 25 °C. Initial [EGCG] = 200 mg L^−1^.

**Figure 4 molecules-25-04249-f004:**
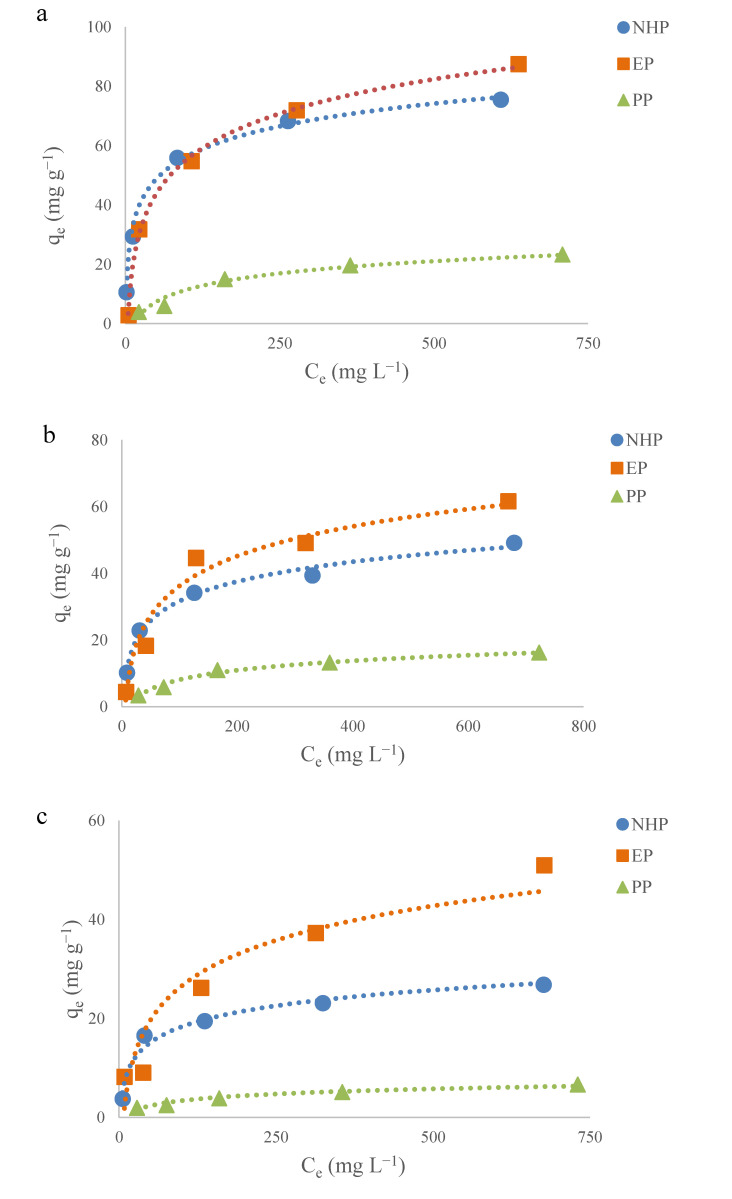
Comparison of equilibrium concentration of EGCG (C_e_) and adsorption capacity (q_e_) between NHP, EP and unprocessed pomelo peel (PP) at 25 (**a**), 40 (**b**) and 55 °C (**c**). The data for unprocessed pomelo peel are from reference [16].

**Figure 5 molecules-25-04249-f005:**
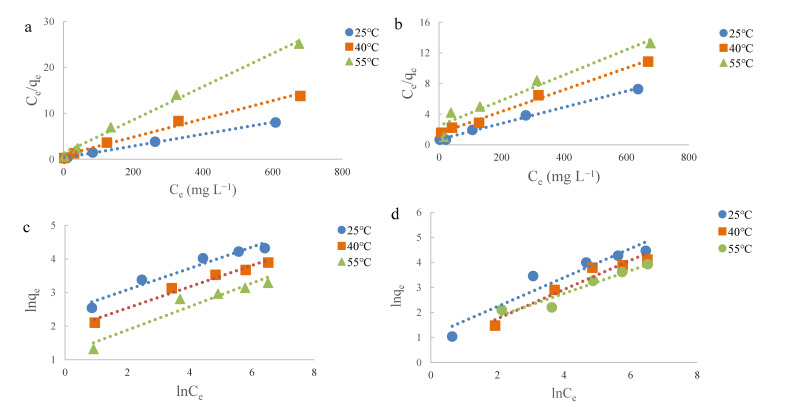
Plots of Langumuir isotherms of EGCG onto NHP (**a**), EP (**b**) and Freundlich isotherms of EGCG onto NHP (**c**) and EP (**d**) at different temperatures.

**Table 1 molecules-25-04249-t001:** Kinetic parameters of EGCG adsorption onto processed pomelo peel *.

C_0_ (mg L^−1^)	Adsorbents	q_e.exp_ (mg g^−1^)	q_e.cal_ (mg g^−1^)	k	R^2^	Relative Error (%)
**Pseudo-first-order model k_1_ (min^−1^)**						
200	NHP	52.83	46.76	0.0075	0.86	11.49
	EP	67.66	62.04	0.0095	0.93	8.31
**Pseudo-second-order model k_2_ (g mg^−1^ min^−1^)**						
200	NHP	52.83	54.64	0.00019	0.98	3.43
	EP	67.66	70.42	0.00021	0.99	4.08

* q_e.exp_, experimental data; q_e.cal_, calculated from the model. Relative error = 100 × (|q_e.cal_ − q_e.exp_|)/q_e.exp_.

**Table 2 molecules-25-04249-t002:** Langmuir isotherm parameters for EGCG adsorption onto processed pomelo peel at different temperatures.

Adsorbents	Parameters	Temperature (°C)		
		25	40	55
NHP	K_L_	43.11	24.01	19.33
	Q_m_ (mg g^−1^)	77.52	50.51	27.62
	R^2^	0.99	0.99	0.99
EP	K_L_	15.49	9.44	5.65
	Q_m_ (mg g^−1^)	94.34	70.42	62.89
	R^2^	0.97	0.99	0.99

**Table 3 molecules-25-04249-t003:** Freundlich isotherm parameters for EGCG adsorption onto processed pomelo peel at different temperatures.

Adsorbents	Paramets	Temperature (°C)		
		25	40	55
NHP	K_F_	11.54	6.71	4.74
	1/n	0.33	0.32	0.28
	R^2^	0.95	0.98	0.95
EP	K_F_	2.92	1.77	1.69
	1/n	0.59	0.58	0.53
	R^2^	0.91	0.94	0.96

**Table 4 molecules-25-04249-t004:** Thermodynamic parameters for adsorption of EGCG onto processed pomelo peel at different temperatures.

Adsorbents	ΔH° (kJ mol^−1^)	ΔS° (J mol^−1^ K^−1^)	ΔG° (kJ mol^−1^)		
			25 °C	40 °C	55 °C
NHP	21.50	147.14	−22.53	−24.23	−26.98
EP	27.31	156.90	−19.48	−21.80	−24.19

**Table 5 molecules-25-04249-t005:** Desorption of EGCG from the pretreated pomelo peel by three-stage elution *.

Sample	Adsorption Amount (mg)	Desorption Amount in the First Elution (mg)	Desorption Amount in the Second Elution (mg)	Desorption Amount in the Third Elution (mg)	Total
NHP	128.65 ± 3.87	1.78 ± 0.14 (1.38%)	103.03 ± 2.44 (80.09%)	5.53 ± 0.34 (4.30%)	110.35 ± 2.49 (85.77%)
EP	140.08 ± 6.09	1.80 ± 0.17 (1.28%)	108.75 ± 1.79 (77.63%)	7.65 ± 0.15 (5.46%)	118.19 ± 1.86 (84.37%)

* The data are shown as mean ± standard deviation of triplicate tests, and the desorption rates are presented in parentheses.

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
