# Peer review of "The Physiochemical Properties and Adsorption Characteristics of Processed Pomelo Peel as a Carrier for Epigallocatechin-3-Gallate"

_molecules, 2020, doi:10.3390/molecules25184249_

Round 1
Reviewer 1 Report
I have reviewed the manuscript entitled: The physiochemical properties and adsorption characteristics of processed pomelo peel as a carrier for epigallocatechin-3-gallate.
In my opinion, the manuscript needs some major modifications, to make the work acceptable for this journal.
Remarks
- Introduction is too short and also not very informative. It must be improved.
- The authors used the FTIR specrtoscopy to show the alteration in the functional groups of pomelo peel samples with different modifications, but it is also very important to show the adsorption of the epigallocatechin-3-gallate on the surface of modified pomelo peel, which is the work objective. Thus, the authors must add the spectra of epigallocatechin-3-gallate only and those of pomelo peel samples (NHP and EP) after adsorption.
- The authors used the SEM to compare the size and the morphology of NHP and EP. Thus, the same scale must be used for the compared samples (Figure 2, between a and c, and b and d).
- In the lines 184-186: Based on the characteristics analysis of these two pomelo peel samples, it is speculated that the EP sample may have a better adsorption capacity than NHP because the former has more specific surface area and potential binding sites, possibly leading to different performance in EGCG delivery. The micrographs obtained by SEM and the Particle size distribution obtained by laser diffraction analyzer can not give the exact information about the specific surface area. The authors need to do the BET analysis for the two samples.
- Finally, the authors must add a part concerning the desorption study of EGCG from the adsorbent surface.
Author Response
Reviewer 1
I have reviewed the manuscript entitled: The physiochemical properties and adsorption characteristics of processed pomelo peel as a carrier for epigallocatechin-3-gallate.
In my opinion, the manuscript needs some major modifications, to make the work acceptable for this journal.
Remarks
- Introduction is too short and also not very informative. It must be improved.
The Introduction section has been re-written in the manuscript. Thanks!
- The authors used the FTIR specrtoscopy to show the alteration in the functional groups of pomelo peel samples with different modifications, but it is also very important to show the adsorption of the epigallocatechin-3-gallate on the surface of modified pomelo peel, which is the work objective. Thus, the authors must add the spectra of epigallocatechin-3-gallate only and those of pomelo peel samples (NHP and EP) after adsorption.
Thank you for your advice. The FTIR spectra of unmodified pomelo peel (PP), epigallocatechin-3-gallate only, and NHP, EP after adsorption have been added in Figure 1.
- The authors used the SEM to compare the size and the morphology of NHP and EP. Thus, the same scale must be used for the compared samples (Figure 2, between a and c, and b and d).
We have supplied SEM photographs of the unmodified pomelo peel (PP), NHP and EP in Figure 2. We have chosen the photographs with closed scale since the SEM observation on PP and NPH, EP samples were from different test batches.
- In the lines 184-186: Based on the characteristics analysis of these two pomelo peel samples, it is speculated that the EP sample may have a better adsorption capacity than NHP because the former has more specific surface area and potential binding sites, possibly leading to different performance in EGCG delivery. The micrographs obtained by SEM and the Particle size distribution obtained by laser diffraction analyzer can not give the exact information about the specific surface area. The authors need to do the BET analysis for the two samples.
This suggestion is constructive, however, we don’t have the facilities for the BET analysis, and the outsourced analytical approach is not available for now since the covid-19 pandemic and the delay of the re-open of the university. The BET analysis can be part of our further study in future.
- Finally, the authors must add a part concerning the desorption study of EGCG from the adsorbent surface.
Thank you for your advice, we have added the result of desorption test in the manuscript.
Submission Date
02 August 2020
Reviewer 2 Report
Dear Author
I consider that the document is interesting, and I think that the writing in some sections needs grammar revision.
I suggest that the authors should include in Figure 1 the FTIR of the unmodified material for better visualization of the changes before and after the modification. As well as in figure 2, the SEM of the three samples and highlight their physical characteristics: pore size, surface appearance….
Line 200
Moreover, Figure 3a demonstrated that the adsorption capacity of EGCG onto EP (67.66 mg g-1) was better than that of NHP (52.83 mg g-1), which was likely caused by the difference in the internal microstructures of the two samples.
How does this discussion go if the characteristics of the material are not shown before the modification? Could this change be attributed to an increase in active sites (which active sites?) resulting from the modification?
I suggest that if it is possible to establish a mechanism for modification and sorption of the material, the authors should attach it to the document.
Author Response
Reviewer 2
Dear Author
I consider that the document is interesting, and I think that the writing in some sections needs grammar revision.
I suggest that the authors should include in Figure 1 the FTIR of the unmodified material for better visualization of the changes before and after the modification. As well as in figure 2, the SEM of the three samples and highlight their physical characteristics: pore size, surface appearance….
Thank you for your advice. The FTIR spectra of unmodified pomelo peel (PP), epigallocatechin-3-gallate only, and NHP, EP after adsorption have been added in Figure 1. We also have supplied SEM photographs of the unmodified pomelo peel (PP), NHP and EP in Figure 2. We have chosen the photographs with closed scale, although the SEM observation on PP and NPH, EP samples were from different test batches.
Line 200
Moreover, Figure 3a demonstrated that the adsorption capacity of EGCG onto EP (67.66 mg g-1) was better than that of NHP (52.83 mg g-1), which was likely caused by the difference in the internal microstructures of the two samples.How does this discussion go if the characteristics of the material are not shown before the modification? Could this change be attributed to an increase in active sites (which active sites?) resulting from the modification?
I suggest that if it is possible to establish a mechanism for modification and sorption of the material, the authors should attach it to the document.
Thank you for your advice, we have added the section in the manuscript to discuss the relationship between the modification and adsorption capacity improvement of the peel samples.
Submission Date
02 August 2020
Reviewer 3 Report
The manuscript titled “The physiochemical properties and adsorption characteristics of processed pomelo peel as a carrier for epigallocatechin-3-gallate” presents an interesting study on the adsorption process of epigallocatechin-3-gallate on modified pomelo peel as function of the process modification. The experiments and results are discussed accordingly and sound. However, some comments and suggestions came up after revision.
- In the introduction, authors mention that stability and bioavailability of epigallocatechin-3-gallate improved after encapsulation. Can the authors add some references about the immobilization of epigallocatechin-3-gallate and the effect on stability and bioavailability?
- In the FTIR discussion, add the spectrum for the unmodified pomelo peel to better understanding of changes in the EP sample.
- What kind of compounds can be removed from pomelo peel with ethanol treatment, accordingly with pomelo composition? Can be detected using FTIR analysis?
- In FTIR discussion, authors only mention oxidation of pomelo peel and the presence of carboxylic groups in NHP sample, but Can mention what kind of compounds were removed after alkaline-acid treatment accordingly with pomelo composition?
- Did the authors study the adsorption process at different pH to select 3.8-4.0 to set the experiments? Can the authors explain why an acidic medium was selected for the adsorption studies?
- In the adsorption kinetics discussion, the authors said “Figure 3a, demonstrated that the adsorption capacity …” Can the authors change the term to “the equilibrium adsorption capacity”?
- From the kinetics study authors conclude that the adsorption process is by chemisorption due to the valence forces; these are due to electrostatic interactions? how occur in this system? In addition, the thermodynamic study suggests that the process is mainly by physisorption mechanism. Add an explanation addressing the difference (inconsistence).
- In the adsorption study at different temperature, the adsorption capacity decrease with the temperature, and authors explained that EGCG escape from adsorbent surface. Can the authors add a better explanation about the role of temperature in the adsorption including the chemisorption/physisorption process? In addition, EGCG solubility change with the temperature? Is there a role of the solubility on the adsorption process?
- From the Freundlich isotherm parameters, 1/n is not close to 1, for any case, the highest value is 0.59, How can the authors consider a homogeneous adsorption layer?
- In the conclusion is important to address the differences in the modification of pomelo peel, and the effect on adsorption properties. Authors just mention that modification increases the adsorption of unmodified, but the manuscript discusses two different modified materials.
- In the results and conclusion authors should address the modification process due to alkaline-acidic process requires several steps while modification with ethanol is easier leading in a better adsorption capacity, but in the discussion, authors didn’t address the reason.
Author Response
Reviewer 3
The manuscript titled “The physiochemical properties and adsorption characteristics of processed pomelo peel as a carrier for epigallocatechin-3-gallate” presents an interesting study on the adsorption process of epigallocatechin-3-gallate on modified pomelo peel as function of the process modification. The experiments and results are discussed accordingly and sound. However, some comments and suggestions came up after revision.
1. In the introduction, authors mention that stability and bioavailability of epigallocatechin-3-gallate improved after encapsulation. Can the authors add some references about the immobilization of epigallocatechin-3-gallate and the effect on stability and bioavailability?
Thank you for your advice! We have added the results of some references about the immobilization of epigallocatechin-3-gallate and the effect on stability and bioavailability.
2. In the FTIR discussion, add the spectrum for the unmodified pomelo peel to better understanding of changes in the EP sample.
Thank you for your suggestion. The FTIR spectra of unmodified pomelo peel (PP), epigallocatechin-3-gallate only, and NHP, EP after adsorption have been added in Figure 1.
3. What kind of compounds can be removed from pomelo peel with ethanol treatment, accordingly with pomelo composition? Can be detected using FTIR analysis?
According to our previous studies by using FTIR spectra (Esterification of sugarcane bagasse by citric acid for Pb2+ adsorption: effect of different chemical pretreatment methods, Chemical Pretreatment of Rice Straw Biochar: Efect on Biochar Properties and Hexavalent Chromium Adsorption, Effect of Pretreatment on the Adsorption Performance of Ni/ZnO Adsorbent for Dibenzothiophene Desulfurization ect.), the treatment of ethanol elution would remove some organic compounds in lignocellulose materials, to enhance the adsorption capacities of biosorbents. The functional groups of these removed compounds could be detected and postulated by the comparison of FTIR spectra before and after modification. We have added some statements on the removed compounds by using the two pretreatments.
4. In FTIR discussion, authors only mention oxidation of pomelo peel and the presence of carboxylic groups in NHP sample, but Can mention what kind of compounds were removed after alkaline-acid treatment accordingly with pomelo composition?
We have re-written the statement in the manuscript.
5. Did the authors study the adsorption process at different pH to select 3.8-4.0 to set the experiments? Can the authors explain why an acidic medium was selected for the adsorption studies?
The pH value of the adsorption system was around 3.8-4.0, without any adjust by adding acid or alkaline. This environmental pH is suitable for the stability of EGCG with little isomerization caused by the adverse pH (pH>6) according to our previous study (Protection of Epigallocatechin Gallate against Degradation during in Vitro Digestion Using Apple Pomace as a Carrier).
6. In the adsorption kinetics discussion, the authors said “Figure 3a, demonstrated that the adsorption capacity …” Can the authors change the term to “the equilibrium adsorption capacity”?
Thank you for your advice, we have made the revision in the manuscript.
7. From the kinetics study authors conclude that the adsorption process is by chemisorption due to the valence forces; these are due to electrostatic interactions? how occur in this system? In addition, the thermodynamic study suggests that the process is mainly by physisorption mechanism. Add an explanation addressing the difference (inconsistence).
As we have revised in the manuscript, it’s inferred that hydrogen bonds formed among hydroxyl groups in galloyl group of EGCG and oxygen atoms of the glycosidic linkages in polysaccharides and the hydrophobic interactions between the benzene rings or alkanes of lignocellulose and phenolic rings of EGCG should be the major chemisorption form during the adsorption process of EGCG onto peel samples, although Van der Waals interaction are also likely to be involved in the interaction.
We have re-written the statement and added the explanation in section 3.4.
8. In the adsorption study at different temperature, the adsorption capacity decrease with the temperature, and authors explained that EGCG escape from adsorbent surface. Can the authors add a better explanation about the role of temperature in the adsorption including the chemisorption/physisorption process? In addition, EGCG solubility change with the temperature? Is there a role of the solubility on the adsorption process?
Thank you for your advice. We have added some explanations to the impact of temperature on chemisorption/physisorption process in the manuscript. The solubility of EGCG is affected by the temperature indeed. The solubility would be increasing along with the elevating of temperature in certain range. However, in case of heating of catechins solution, concentrations of EGCG, ECG, EGC, EC and total catechins were all significantly decreased attributing to the epimerization according to the previous study (Tea extraction methods in relation to control of epimerization of tea catechins). To date, the systematic study on the role of solubility on the EGCG adsorption process is rare, this could be a focus in future study.
9. From the Freundlich isotherm parameters, 1/n is not close to 1, for any case, the highest value is 0.59, How can the authors consider a homogeneous adsorption layer?
We have revised this part in the manuscript.
10. In the conclusion is important to address the differences in the modification of pomelo peel, and the effect on adsorption properties. Authors just mention that modification increases the adsorption of unmodified, but the manuscript discusses two different modified materials.
Thank you for your advice, we have made the revision in the manuscript.
11. In the results and conclusion authors should address the modification process due to alkaline-acidic process requires several steps while modification with ethanol is easier leading in a better adsorption capacity, but in the discussion, authors didn’t address the reason.
We have supplied the explanation on the better adsorption capacity of EP peel sample.
Reviewer 4 Report
Review on the manuscript entitled “The physiochemical properties and adsorption characteristics of processed pomelo peel as a carrier for epigallocatechin-3-gallate”, authors Liangyu Wu, Guoying Zhang and Jinke Lin.
The manuscript is well written but I consider it could use some improvements.
Please consider the following queries:
- How can you control the quality of the pomelo peel? What if the fruit is not ripe yet, how can affect the properties?
- I understood that the pre-treatment (alkaline, acid, Et-OH) could improve the adsorption capacity, but the freezing and storing at -20 °C could also influence the porosity/surface?
- On FTIR spectra the Transmittance is in % but no scale in provided, should be used arbitrary units.
- Can you explain what type of chemical bonds are developed between the un/treated pomelo peel and EGCG? (Beside the available sites on the surface of the peel that are responsible for the adsorption)
- What can you tell about the desorption process? How can be the EGCG released? Have you tried a controlled release of the EGCG?
I think that the desorption study, is mandatory in this case and will provide vital information for pomelo peel as a carrier for EGCG
Author Response
Reviewer 4
Review on the manuscript entitled “The physiochemical properties and adsorption characteristics of processed pomelo peel as a carrier for epigallocatechin-3-gallate”, authors Liangyu Wu, Guoying Zhang and Jinke Lin.
The manuscript is well written but I consider it could use some improvements.
Please consider the following queries:
- How can you control the quality of the pomelo peel? What if the fruit is not ripe yet, how can affect the properties?
The pomelo is a commonly fresh-eaten fruit in China, most pomelo fruits are subjected to fresh-eating or juicing process in food industry. In this study, the peel samples were collected from 15-20 ripen pomelos according to the commercial standard (yellow peel, sweet flesh…) in a local market in Fuzhou city, then we homogenized the peel by a colloid mill to control the quality of the peel.
The fiber contents (cellulose, hemicellulose, lignin, some few pectin) of unripen peel samples might be different from that of ripen fruit, although we have never used the commercial standard unripen pomele for the peel sample.
- I understood that the pre-treatment (alkaline, acid, Et-OH) could improve the adsorption capacity, but the freezing and storing at -20 °C could also influence the porosity/surface?
According to our previous studies on the absorbents stored at -20 and 4 ℃ (The batch adsorption of the epigallocatechin gallate onto apple pomace, Application of NaOH-HCl-Modified Apple Pomace to Binding Epigallocatechin Gallate), the freezing and storing at -20 ℃ had negligible effect on the porosity of adsorbents, the improvements in adsorption properties are generally due to the elution pretreatments. Moreover, the peel samples stored at -20 ℃ were unfrozen until to room temperature, then the peel samples were rinsed in Milli-Q water prior to further analysis to eliminate interference.
- On FTIR spectra the Transmittance is in % but no scale in provided, should be used arbitrary units.
Thank you for your suggestion. The scale in FTIR spectra ranged from 0 – 100 %. The scale range in FTIR figure was removed in the manuscript since we combined several FTIR spectra from different samples into one coordinate, therefore we just removed the scale for a better exhibition.
- Can you explain what type of chemical bonds are developed between the un/treated pomelo peel and EGCG? (Beside the available sites on the surface of the peel that are responsible for the adsorption)
It is showed that polyphenols could be adsorbed by plant-based biosorbents. Since the most components of the plant-based biosorbents are dietary fibers, the interactions between EGCG and pomelo peel are hydrogen bond and hydrophobic interactions according to several research and reviewing papers (Binding of polyphenols to plant cell wall analogues – Part 1:Anthocyanins, Binding of polyphenols to plant cell wall analogues – Part 2: Phenolic acids, Structural characterization of inclusion complexes between cyanidin-3-O-glucoside and beta-cyclodextrin, Interactions of polyphenols with carbohydrates, lipids and proteins). Hydrogen bonds are probably formed between hydroxyl groups of EGCG and oxygen atoms of the glycosidic linkages of polysaccharides (cellulose, pectin, dietary fibers etc.). The benzene rings or alkanes of lignocellulose could potentially interact with phenolic rings of tea catechins to form hydrophobic interaction, which also result in the chemical adsorption between pomelo peel and EGCG.
- What can you tell about the desorption process? How can be the EGCG released? Have you tried a controlled release of the EGCG?
I think that the desorption study, is mandatory in this case and will provide vital information for pomelo peel as a carrier for EGCG
We have added the result of desorption test in the revised manuscript. In our previous studies (Protection of Epigallocatechin Gallate against Degradation during in Vitro Digestion Using Apple Pomace as a Carrier, Development of Phyllanthus emblica (L) fruit as a carrier for EGCG: Interaction and in vitro digestion study), the release of EGCG from EGCG-adsorbent complex have been evaluated by employing a sequential in vitro gastrointestinal digestion, including gastric (120 min) and intestinal (180 min) digestion simulation. It’s showed the binding between the EGCG and adsorbents were degraded gradually under the action of enzymes (pepsin and pancreatin) and adverse environmental pH (>6.0). Most of the EGCG were released from the adsorbents and decomposed during the intestinal digestion, and the main factor influenced the degradation of EGCG was the environmental pH.
Submission Date
02 August 2020
Date of this review
18 Aug 2020 13:52:30
Round 2
Reviewer 1 Report
Dear Authors,
The revision you have made is satisfactory.
Thank you.
Reviewer 4 Report
The manuscript it suitable for publication in its present form.